# Carbon Nano-Onions–Polyvinyl Alcohol Nanocomposite for Resistive Monitoring of Relative Humidity

**DOI:** 10.3390/s25103047

**Published:** 2025-05-12

**Authors:** Bogdan-Catalin Serban, Niculae Dumbravescu, Octavian Buiu, Marius Bumbac, Carmen Dumbravescu, Mihai Brezeanu, Cristina Pachiu, Cristina-Mihaela Nicolescu, Cosmin Romanitan, Oana Brincoveanu

**Affiliations:** 1National Institute for Research and Development in Microtechnologies, IMT-Bucharest, Str Erou Iancu Nicolae 126A, 077190 Voluntari, Romania; niculae.dumbravescu@imt.ro (N.D.); octavian.buiu@imt.ro (O.B.); cristina.pachiu@imt.ro (C.P.); cosmin.romanitan@imt.ro (C.R.); oana.brincoveanu@imt.ro (O.B.); 2Sciences and Advanced Technologies Department, Faculty of Sciences and Arts, Valahia University of Târgoviște, Aleea Sinaia nr 13, 130004 Targoviste, Romania; 3Institute of Multidisciplinary Research for Science Technology, Valahia University of Târgoviște, Aleea Sinaia 13, 130004 Târgoviște, Romania; cristina.nicolescu@valahia.ro; 4MIGSO-PCUBED, The Light One Building, Bd Iuliu Maniu 6Q, 061344 Bucuresti, Romania; carmen.dumbravescu-extern@renault.com; 5Faculty of Electronics, Telecommunications, and IT, National University of Science and Technology Politehnica Bucharest, Bd Iuliu Maniu 1-3, 061071 Bucharest, Romania; mbrezeanu@upb.ro

**Keywords:** carbon nano-onions, polyvinyl alcohol, resistive relative humidity sensor, swelling

## Abstract

**Highlights:**

**What are the main findings?**

**What is the implication of the main finding?**

**Abstract:**

This paper reports several preliminary investigations concerning the relative humidity (RH) detection response of a chemiresistive sensor that uses a novel sensing layer based on pristine carbon nano-onions (CNOs) and polyvinyl alcohol (PVA) at a 1/1 and 2/1 *w*/*w* ratio. The sensing device, including a Si/SiO_2_ substrate and gold electrodes, is obtained by depositing the CNOs–PVA aqueous suspension on the sensing structure by drop casting. The composition and morphology of the sensing film are explored by means of scanning electron microscopy, Raman spectroscopy, atomic force microscopy, and X-ray diffraction. The manufactured sensor’s room temperature RH detection performance is examined by applying a continuous flow of the electric current between the interdigitated electrodes and measuring the voltage as the RH varies from 5% to 95%. For RH below 82% (sensing layer based on CNOs–PVA at 1/1 *w*/*w* ratio) or below 50.5% (sensing layer based on CNOs–PVA at 2/1 *w*/*w* ratio), the resistance varies linearly with RH, with a moderate slope. The newly developed sensor, using CNOs–PVA at a 1:1 ratio (*w*/*w*), responded as well as or better than the reference sensor. At the same time, the recorded recovery time was about 30 s, which is half the recovery time of the reference sensor. Additionally, the changes in resistance (ΔR/ΔRH) for different humidity levels showed that the CNOs–PVA layer at 1:1 was more sensitive at humidity levels above 80%. The main RH sensing mechanisms considered and discussed are the decrease in the hole concentration in the CNOs during the interaction with an electron donor molecule, such as water, and the swelling of the hydrophilic PVA. The experimental RH detection data are analyzed and compared with the RH sensing results reported in previously published work on RH detectors employing sensing layers based on oxidized carbon nanohorns–polyvinylpirrolidone (PVP), oxidized carbon nanohorns–PVA and CNOs–polyvinylpyrrolidone.

## 1. Introduction

In recent decades, relative humidity (RH) sensors have earned a lot of attention due to their various applications in areas such as buildings ventilation control, health monitoring (biomedical analysis), food and beverage storage, cosmetics, agriculture, pharmaceuticals, microelectronics (clean rooms for manufacturing and testing electronic devices), automotive (cabin moisture control), weather stations, soft robotics, and the paper industry, etc. [1,2]. The global RH sensors market was valued at USD 1.41 billion in 2024, and is anticipated to grow continuously in the coming years [3].

Commercially available RH sensors must meet features such as linearity, high sensitivity, reproducibility, low hysteresis, fast response time, long-term stability, simple design, small size, and low cost [4]. Typically, RH sensors include three main components: a rigid (ceramic, silicone, etc.) or flexible (polyimide, polyethylene terephthalate) substrate, a sensitive material, and peripheral electronic circuits [5]. Among various types of humidity sensors, resistance-based sensors, such as the one developed in this study, are particularly noteworthy due to their simplicity, cost-effectiveness, and high sensitivity, which make them highly suitable for a wide range of applications. While capacitive and impedance-based humidity sensors also feature simple structures and are commercially available, resistance-based sensors offer unique advantages, including ease of fabrication and the potential for integration into low-power and miniaturized systems. These characteristics position resistance-based sensors as a promising alternative in humidity sensing technologies [6].

Recently, numerous types of materials have been explored as potential RH sensing layers: metal oxide semiconductors [7,8], polyelectrolytes [9,10], and perovskites [11,12]. Carbon-based sensing layers were also employed as RH sensing layers: carbon dots [13], amorphous carbon [14], hydrogenated amorphous carbon [15], carbon nanofibers [16], nanodiamonds [17], fullerenes [18], graphene [19], graphene oxide [20], reduced graphene oxide [21], carbon nanohorns (CNHs), their nanocomposites and nanohybrids [22,23,24], oxidized carbon nano-onions (CNOs), and their nanocomposites [25].

Carbon nano-onions are multi-layered, concentric, spherical structures composed of graphene-like shells. They resemble “onions”, with each layer being a curved carbon sheet, typically arranged around a central core (often a fullerene or amorphous carbon). CNOs are typically in the nanometer range, with diameters ranging from a few nanometers to tens of nanometers, depending on the synthesis method and application. Carbon nano-onions (CNOs) are a distinct and highly specialized form of carbon nanomaterial with unique structural and functional properties [25].

At the same time, due to their low-cost synthesis, large-scale manufacturing capability, versatile functionalization, low density, structural flexibility, and facile deposition on different substrates, polymers have also received significant attention in recent decades as RH sensing layers. Organic polymers used as hygroscopic materials in RH sensing films include hydrophobic polymers, such as fluorinated polyimide [26], poly (methyl methacrylate) (PMMA) [27], and cellulose acetate butyrate [28]; conductive polymers, such as polythiophene [29] and polyaniline [30]; polymer electrolytes [31]; and hydrophilic polymers, like gelatin [32], polyvinylpyrrolidone (PVP), and polyvinyl alcohol (PVA) [33]. Finally, biopolymers have also gained increased interest for being used within RH sensing layers due to their excellent biodegradability and non-toxicity. Examples of such materials include polylactic/glycolic acid [34], chitosan [35], polyglutamic acid, and polylysine [36].

However, RH sensing layers based only on polymers typically exhibit drawbacks, such as low RH sensitivity, substantial backbone mechanical dissolution, restricted temperature range, and long response time [37]. Consequently, many polymer-based nanocomposites and nanohybrids have been designed, focusing on optimizing the physico-chemical properties of polymers and improving their water molecules detection capability. Polymer–metal nanohybrids [38], polymer–polymer nanocomposites [39], metal oxide–semiconducting polymers [40], metal–organic framework polymers [41], polymer nanofibers, oxide nanofibers, and composite nanofibers [42] are just a few examples of nanocomposite materials that have been used as sensing layers for RH monitoring.

At the same time, nanocomposites and nanohybrids, including polymers and nanocarbon materials, have also been demonstrated as RH sensing films [43].

Several examples of such nanocomposites and nanohybrids and their associated sensing technologies are presented in Table 1. These sensor types exhibit unique strengths and limitations depending on their application scenarios. Among these, resistance-based humidity sensors, such as the one developed in this work, stand out due to their simplicity, cost-effectiveness, and high sensitivity, making them suitable for various applications. Furthermore, carbon-based humidity sensors typically exhibit positive resistance characteristics due to the hygroscopic swelling effect of carbon/organic composite materials. This phenomenon, which causes an increase in resistance with rising humidity, is particularly significant for carbon/organic materials, as their swelling disrupts conductive pathways. Recent progress in carbon-based humidity sensors highlights the potential of these materials in developing highly sensitive and reliable humidity sensors [14].

This study presents the synthesis and characterization of RH sensing films based on a nanocomposite comprising CNOs and PVA at 1:1 and 2:1 *w*/*w* ratios. The room temperature (RT) RH sensing response of a chemiresistive sensor using the RH-synthesized sensing layer is investigated. This is the first time a nanocomposite based on CNOs and PVA is reportedly used for RH sensing. The experimental RH detection data are analyzed and compared with the previously published RH sensing results measured on RH detectors employing nanocomposites comprising CNHox–PVP, CNHox–PVA, and MWCNTs–PVA as sensing layers.

## 2. Materials and Methods

### 2.1. Materials

CNOs (Figure 1a) were procured from Shanghai Epoch Material Co., Ltd., while PVA powder (87–90% hydrolyzed, average mol wt. 30,000–70,000—(Figure 1b)) was acquired from Sigma Aldrich (Bucharest, Romania). The chemicals were of the highest available grade and were used as received without additional purification.

### 2.2. Synthesis of the Nanocarbon Composite Sensing Layer and Experimental Setup

The synthesis of the sensitive film based on the CNOs–PVA nanocomposite at a 1/1 mass ratio started with preparing the PVA solution. The hydrophilic polymer solution was prepared by gradually dissolving 6 mg of PVA in 12 mL of deionized water for 6 h at 90 °C in an ultrasonic bath (FS20D Fisher Scientific, Schwerte, Germany), working at 42 kHz (70 W output power). CNOs (6 mg) were dispersed in a clear water-based PVA solution and subjected to intensive stirring in an ultrasonic bath for 6 h, maintaining the temperature at 50–60 °C. The dispersion was drop-cast over a metallic interdigitated (IDT) structure while the contact area was masked [60]. The IDT dual-comb structure was manufactured on a Si substrate (470 µm thickness), covered by a SiO_2_ layer (1 µm thickness) (Figure 2). The sensing film was heated at 70 °C for 2 h in vacuum, and, finally, the whole RH sensing device was dried at 70 °C for 2 h [61]. The synthesis of the sensitive film based on the CNOs–PVA nanocomposite at a 2/1 mass ratio was performed similarly, except for the CNO/PVA mass ratio.

To assess its RH monitoring performance, the chemiresistive manufactured sensor (MS), utilizing, for the first time, the CNOs–PVA nanocomposite (in 1:1 and 2:1 *w*/*w* ratios) as RH sensing film, was tested in parallel with a capacitive, commercially available, RH sensor (REF), which acted as a reference device. Both devices were placed inside a testing box depicted in Figure 3.

The experimental setup employing two bubblers was selected based on the results of preliminary tests conducted to evaluate the effectiveness of using one or multiple bubblers. These tests aimed to determine whether the humidity level could remain stable when the gas flow was maintained at a constant rate for approximately 300 s. The findings indicated that the configuration with two bubblers provided the most consistent humidity levels under these conditions. In the proposed experimental setup, it was observed that achieving a stable response time for the reference sensor during a humidity transition from near 0% RH to approximately 10% RH was not possible. This limitation was not related to the performance of the reference or tested sensor, but rather to the inability to reproducibly control the humidity increase within the 0–10% RH range in the mixing chamber. Consequently, the response time under these conditions was deemed unreliable.

In the proposed experimental framework, dry nitrogen was purged through bubblers containing demineralized water to achieve RH values within the 5–95% range. Relative humidity sensing measurements were performed using a specialized testing bench (Figure 3). To vary the relative humidity (RH) in the testing chamber, dry nitrogen was passed through two containers in series containing deionized water. The humidity level was adjusted by mixing dry and humidified nitrogen in different ratios. In the mixing chamber (indicated by the green square in Figure 3), the gases from both paths formed a homogeneous mixture, which was then directed into the testing chamber. The chamber housed two sensors: our resistive sensing structure (i.e., MS-measured sensor), which uses GO/CNHox/PVP as the sensing layer, and a commercially available capacitive relative humidity sensor (referred to as REF—reference sensor). The commercial sensor was used to validate the relative humidity levels indicated by the mass flow controller (MFC) system. Both sensors were positioned close to each other and near the gas inlet, ensuring identical exposure to the gas flow (dry nitrogen) and consistent experimental conditions for reliable conclusions [56,59].

Before starting the experiments, the devices were exposed to dry nitrogen for 6 h to achieve an almost moisture-free environment. MS and REF were positioned near the gas inlet to be subject to quasi-identical testing environments. Current variations were induced using a Keithley 6620 current source. The output voltage was recorded, and the electrical resistance was computed with a PicoLog data logger. The total electrical power consumption of MS was less than 2 mW, its low power consumption being a significant advantage of this type of sensor. 

## 3. Results

### 3.1. Surface Topography

The surface topography of the sensing films based on the CNOs–PVA (carbon nano-onions–polyvinyl alcohol) nanocomposite was characterized using scanning electron microscopy (SEM). Figure 4 presents the morphology of the sensing layer prepared with a CNO–PVA ratio of 1:1 (*w*/*w*). As shown in Figure 4a, the surface of the film is generally homogeneous, with a good distribution of nanocarbon materials throughout the polymer matrix. However, closer inspection (Figure 4b) reveals the presence of localized agglomerations, forming isolated islands of CNO nanoparticles and distinct PVA polymer clusters.

The CNOs were dispersed into a transparent water-based PVA solution and subjected to intensive ultrasonic treatment for 6 h at a controlled temperature of 50–60 °C. These preparation conditions were sufficient to promote a relatively good dispersion of the CNOs within the PVA matrix. After deposition, the sensing films were annealed under moderate vacuum conditions at a moderate temperature for 4 h to minimize nanoparticle agglomeration. Despite these careful processing steps, some minor aggregation of nanoparticles was still observed, as expected in such nanocomposites.

Overall, the surface morphology indicates an acceptable level of homogeneity, which is critical for ensuring reliable sensor performance. The good film continuity and dispersion achieved, as shown in Figure 4a, confirm that the prepared CNOs–PVA sensing layer is suitable for use in the intended resistive humidity sensor applications.

### 3.2. Raman Spectroscopy

Raman spectroscopy measurements were carried out at RT using a WiTec Raman spectrometer (Alpha-SNOM 300 S, WiTec GmbH, Ulm, Germany) with an excitation wavelength of 532 nm. A diode-pumped solid-state laser emitting at 532 nm and delivering a maximum power of 145 mW was directed onto the sample through an objective lens with a 6 mm working distance, integrated into a Thorlabs MY100X-806 microscope (Newton, NJ, USA). The laser beam was concentrated into a spot size of nearly 1.0 µm.

The Raman spectra (Figure 5) were acquired with an exposure time of 20 s, and the scattered light was gathered in backscattering geometry using the same objective lens. A 600 grooves/mm diffraction grating was employed for spectral resolution. The Raman system was calibrated using a silicon wafer’s 520 cm^−1^ Raman peak. The entire process, including spectral acquisition, data processing, and analysis, was managed through WiTec Project Five software, version 5.1.

The Raman spectrum of CNOs displays two distinct bands, between 1300 and 1600 cm^−1^, attributed to the D mode (1331 cm^−1^) and G mode (1559 cm^−1^) [62]. These vibrational features are commonly observed in carbon-based materials and provide crucial insights into their structural characteristics and degree of order.

The G mode at 1559 cm^−1^ corresponds to C–C bond stretching vibrations within the sp^2^-hybridized layers of the carbon framework. This band is linked to the E_2_g vibrational mode of sp^2^-hybridized carbon, indicating well-ordered carbon structures. In the CNOs–PVA nanocomposite, the presence of the G band signifies the existence of carbon layers arranged in a spherical pattern.

Conversely, the D mode at 1331 cm^−1^ is associated with structural defects in the sp^2^ carbon lattice. This mode originates from the breathing vibrations of sp^2^ carbon rings and is influenced by defects, impurities, dopants, or functional groups within the material. In CNOs–PVA, the D band intensity is affected by structural imperfections within the nanocomposite matrix. An increase in D band intensity reflects a higher level of disorder in the carbon network.

The D and G bands (ID/IG) intensity ratio in Raman spectra provides a quantitative measure of defect density and structural disorder in carbon-based materials. For pristine carbon nano onions (CNOs), this ratio is ID/IG = 0.558, while for the CNO/PVA composite, it increases significantly to ID/IG = 1.126. This doubling of the ratio indicates a substantial rise in structural defects and disorder, which can be attributed to the incorporation of polyvinyl alcohol (PVA) into the composite matrix. The interaction between PVA and CNOs, involving the formation of covalent and non-covalent bonds, alters the material’s structural arrangement and Raman scattering intensity. These interactions are critical for the sensing behavior of the composite in resistive sensors [63,64]

At lower frequencies, the Raman spectrum exhibits vibrational modes corresponding to polyvinyl alcohol (PVA), specifically C–H, C–O, and C–C bonds. The spectral peak at 950 cm^−1^ is attributed to C–O stretching vibrations, while the band at 570 cm^−1^ corresponds to out-of-plane hydroxyl bending vibrations. Additionally, peaks detected within the 100–300 cm^−1^ range are linked to C–C bond stretching and C–H bending modes.

### 3.3. X-Ray Diffraction Results

The RH sensing layer’s morphology and crystal structure were also explored by employing X-ray diffraction (XRD). In this respect, an X-ray diffractometer with a nominal power of 9 kW (Rigaku SmartLab Studio II version 4.4) was used. A θ/2θ configuration was utilized to analyze the CNOs pristine powder and the CNOs–PVA nanocomposite deposited on the Si substrate. The latter sample was examined at a low incidence angle, i.e., small X-ray penetration depth, to minimize the signal from the Si substrate.

Figure 6 shows the diffractograms for the CNOs powder and the CNOs–PVA nanocomposite deposited on the Si substrate. The characteristic broad feature between 15 and 30°, centered at 23.2°, corresponds to the (0 0 2) plane of the graphitic carbon peak. The broadening in the (0 0 2) peak is typical for the CNOs [65,66]. Additionally, the broad overlapped features around 42° and 45°, attributed to (100) and (110), emphasize the disordered structure of the graphene rings within the CNOs, in agreement with the rather reduced height of the G peak with respect to the D peak from the Raman spectra of CNOs–PVA nanocomposite.

These patterns agree with other studies on the microstructure of CNO-based composites [25]. For CNOs–PVA nanocomposite deposited on a Si substrate, only a low-intensity band is observed, which is correlated with the semi-crystalline nature of pure PVA [67].

### 3.4. Atomic Force Microscopy Measurements

Atomic force microscopy (AFM) analysis (Figure 7 and Figure 8) was conducted using a WITec scanning near-field optical microscope operating in the tapping mode for CNOs–PVA layer with both 1/1 and 2/1 *w*/*w* ratios. The analysis employed a Si_3_N_4_ AFM cantilever with a length of 125 µm, a force constant of 40 N/m, and a frequency of 300 kHz. The surface parameters were calculated using Project FIVE 5.0 Witec software.

The values of the roughness parameters for CNOs–PVA 1/1 *w*/*w* were SDQ (root mean square gradient) = 0.759 nm, SA (arithmetic average) = 353.83 nm, and SQ (root mean square average) = 435.40 nm. The high values of SA and SQ roughness parameters indicate that the surface was not smooth and had a pronounced topography. The moderate value of SDQ suggests that, while the surface was rough, the slopes were not excessively steep, implying that the roughness distribution was relatively uniform rather than jagged or highly irregular. The values for the roughness parameters for CNOs–PVA 2/1 *w*/*w* were SDQ =1.12 nm, SA = 366.35 nm, and SQ = 461.59 nm.

### 3.5. RH Monitoring Capability

The RT RH sensing performance of MS using CNOs–PVA as the sensing layer, at a 1:1 and 2:1 *w*/*w* ratio was explored by applying a continuous electric current between the interdigitated electrodes and measuring the voltage as the RH varied from 5% to 95%.

As depicted in Figure 9, altering the ratio between CNOs and PVA significantly affects the resistance variation with RH in the test chamber. The resistance exhibits continuous fluctuations across a wide range for the sensing layer with a higher CNOs content, even when the humidity levels remain relatively stable.

Significant differences in resistance variation were observed between the two RH sensing layers. The sensor utilizing a CNOs–PVA ratio of 1:1 (*w*/*w*) demonstrated resistance values ranging from 5 to 7.7 kΩ when exposed to relative humidity (RH) levels between 5% and 95%. In contrast, the sensor with a CNOs–PVA ratio of 2:1 (*w*/*w*) exhibited much higher resistance values, varying between 31 and 40 kΩ under the same RH conditions (Figure 9). Although a higher concentration of conductive CNOs is expected to enhance electrical conductivity, the experimental results indicate a resistance increase for the CNO–PVA (2:1 *w*/*w*) sensing layer. This unexpected behavior can be attributed to the agglomeration of nanoparticles at higher CNO concentrations, which disrupts the formation of a continuous conductive network. Instead, isolated clusters with poor electrical connectivity are formed. Furthermore, an excessive CNO content compromises the continuity of the hydrophilic PVA matrix, impairing charge transport and reducing humidity sensing performance. Contact resistance between inadequately connected CNO clusters may also contribute to the observed increase in resistance.

The hysteresis curves in Figure 9 further highlight the differences between the two sensors. The sensor with a CNO–PVA ratio of 1:1 stabilizes after the first operating cycle and exhibits a response comparable to that of a commercial sensor by the third cycle. In contrast, the sensor with a ratio of 2:1 fails to stabilize even after the first two cycles, indicating that higher CNO concentrations are unsuitable for use in sensing devices.

These findings are supported by the data in Figure 10, which demonstrate the excellent stability of the CNO–PVA 1:1 sensor over multiple operating cycles.

The stability of the proposed RH sensing materials was evaluated by measuring the sensitive layer’s resistance over multiple operating cycles for three identically manufactured sensors subjected to repeated RH variations across the 5–95% range. Figure 10 presents the variation with RH of the ratio (Ri − Rf)/Ri. The results indicate that the CNOs–PVA 1:1 (*w*/*w*) sensing layer demonstrates significantly larger stability than the CNOs–PVA 2:1 (*w*/*w*) sensing layer.

The response time and sensitivity analysis were conducted only for the sensor with the CNOs–PVA at 1:1 (*w*/*w*) sensing layer, as presented in Figure 11.

From the graph of the stepwise evolution of the sensor resistance, the response times were calculated according to the following formula:$$R.T.\;\left(response\;time\right)=t_{90\%}-t_{10\%}$$

For each RH step, the sensitivity was calculated as follows:$$S=\frac{∆R}{∆RH}=\frac{\left(R_{i+1}-R_{i}\right)}{\left({RH}_{i+1}-{RH}_{i}\right)}$$

The variation in ΔR/ΔRH for different RH steps indicated that the CNOs–PVA layer at 1:1 (*w*/*w*) exhibited distinct behavior at RH levels above 80%. The t_90%_ and t_10%_ values were determined for each RH step, as shown in Figure 9. The response times of MS ranged from 40 to 100 s, with the longest times recorded when RH decreased to 70%, a trend also observed for REF. The ratio of the response times (Figure 11c) demonstrated that MS employing CNOs–PVA at 1:1 (*w*/*w*) generally performed comparably or better than REF, except for RH between 30% and 40%, where MS was slower than REF for all three testing cycles. The recovery time of MS during the RH drop from 95% to 10% was approximately 30 s, which was half the recovery time of REF (~65 s).

A key characteristic of the sensor employing CNOs–PVA at 1:1 (*w*/*w*) is an inflection point in the evolution of the (Rf − Ri)/Ri factor in the 80–85% RH domain, as shown in Figure 12. This behavior is observed during both the water molecules sorption and desorption stages.

RH monitoring was performed during successive operating cycles for the resistive RH sensors employing the layers based on CNOs–PVA with a 1:1 (*w*/*w*) ratio (Figure 12a). The results emphasize that the resistance of MS increased with RH over the entire RH range considered. For RH below 85%, the resistance variation with RH was linear, with a moderate slope. The slope became significantly steeper for RH values above the 85% threshold; thus, a significantly higher sensitivity was measured. The response of MS was reproducible, with the variation in the resistance and the differences between the sorption and desorption curves being around ±5%.

In the case of the CNOs–PVA at 2:1 (*w*/*w*) sensing layer, RH monitoring was performed for 10 operating cycles (Figure 12b). In this case, the sensing layer stabilized significantly slower compared to the CNOs–PVA at 1:1 (*w*/*w*) sensing layer. The adsorption curve approached the desorption curve only after 8 adsorption–desorption cycles. At the same time, the inflection point during adsorption showed at approximately RH = 50.5% (black line), compared to the one for desorption, which occurred at approximately RH = 56.5% (blue line) (Figure 12b). The inflection points are determined only for operating cycles 9 (red square) and 10 (blue square) since during the first eight cycles, there are too large differences between the desorption and adsorption curves and between cycles. A possible explanation for the much significantly slower stabilization of the sensing layer and increased hysteresis in the case of the sensing layer based on CNOs–PVA for a 2:1 (*w*/*w*) ratio may rely on the fact that an increased percent of nanocarbon material in the matrix nanocomposite yielded a less homogeneous RH sensing layer, with more “islands” of agglomerated nanocarbon particles surrounded by hydrophilic polymer.

The lower values of the electrical resistance measured at RT for the RH detector based on the sensing layer employing CNOs–PVA at a 2:1 *w*/*w* ratio compared to the device using CNOs–PVA at 1:1 *w*/*w* ratio can also be explained by the higher percentage of conductive nanocarbon material. For both sensing films (CNOs–PVA 2/1 and 1/1 (*w*/*w*) ratio, respectively), the concentration of CNOs is above the percolation threshold of the nanocarbon material within the hydrophilic PVA matrix. The percolation threshold is the critical concentration of conductive nanomaterials needed to form continuous, electrically conductive pathways throughout the matrix. When the CNOs concentration is above this threshold, electrically percolating paths are established between the two metal electrodes of the proposed RH sensor, allowing the device to effectively measure changes in electrical resistance. Below the percolation threshold, these conductive pathways are not formed, and the sensing device cannot detect any changes in resistance, regardless of the humidity level in the testing environment. This is because the lack of interconnected conductive paths prevents the flow of electrical current, rendering the device insensitive to variations in relative humidity.

## 4. Discussion

The experimental RH detection capabilities of the proposed MS structure can be understood by considering several distinct RH sensing mechanisms.

The first RH detection mechanism relies on CNOs being p-type semiconductors [68]. Once the contact between CNOs and water molecules occurs, the latter donate their electron pair, thus decreasing the concentration of holes in the CNOs structure. As the RH level increases, the hole concentration in CNOs decreases; hence, the CNOs–PVA nanocomposites become less conductive.

Another RH sensing mechanism is related to the interaction between CNOs and water molecules, which can be explained using the hard–soft acid–base (HSAB) principle [69]. This concept is widely applied in nanotechnology to explain the stability of compounds, reaction mechanisms, and potential interactions [70].

Pristine CNOs possess positively charged carriers, which can be compared to hard acids, while water molecules behave as hard bases due to their electron pairs. According to the HSAB principle, chemical interactions are likely between hard acids and hard bases. In this context, water molecules gather near the nanocarbon material (CNOs), leading to a chemical interaction that influences the sensing mechanism [69,70].

As the relative humidity increases, water molecules accumulate in the vicinity of the CNOs and interact with them. This interaction, combined with the swelling of the hydrophilic PVA matrix, increases electrical resistance. The swelling of the PVA matrix increases the distance between conductive CNOs, while the interaction with water molecules further disrupts the conductive pathways within the material. Together, these effects reduce the sensing film’s overall conductivity, translating the humidity increase into a measurable increase in resistance [71,72].

A third RH sensing mechanism that can be considered is the one proposed by Dhonge et al. [73]. According to these authors, for RH > 27.9%, CNOs may change their semiconducting behavior, switching from p-type to n-type. To explain the phenomenon, the authors proposed the mechanism described by the reactions below:H_2_O(g) → H_2_O^+^(ad) + e^−^(1)H_2_O(g) + O_0_^2−^ + V^−^ → 2OH_o_^−^(ad) + e^−^(2)H_2_O(g) + O_0_^2−^ + V^2−^ → 2OH_o_^−^(ad) + e^−^(3)

V^−^ and V^2−^ represent vacancies that can trap one or two electrons within the CNOs. The electrons generated in the absorption process depicted above neutralize the positive charge carriers from the CNOs. Moreover, according to Dhonge et al., as RH increases, the concentration of electrons increases, leading to a change in CNOs to n-type semiconducting behavior. Should this process occur, a decrease in the sensor resistance while increasing RH should be experimentally measured. However, this type of behavior was not measured for the MS investigated in this study.

A fourth RH sensing mechanism considers the self-ionization of water into protons and hydroxyl anions on the surface of the nanocarbon structure, occurring at high RH levels. These ions produced by the dissociation of water (and proton-hopping from one water molecule to another) may increase the overall electrical conductivity of the thin sensing film. This fourth sensing mechanism seems plausible, but it is clear that proton conduction cannot play a significant role in resistive RH sensing.

Given the above-discussed RH sensing mechanism, lowering CNOs holes’ concentration in interaction with water molecules seems to be the most significant RH sensing mechanism for RH < 82% (when employing CNOs–PVA at 1:1 (*w*/*w*) as RH sensing layer) and for RH < 50% (when employing CNOs–PVA at 2:1 (*w*/*w*) as RH sensing layer).

However, a significantly more substantial decrease in the electrical conductivity of the sensing film with the RH was measured for RH higher than the above-listed threshold values. This might be due to a fifth RH sensing mechanism. PVA is a dielectric polymer with hydrophilic properties. When RH is low, the CNOs–PVA nanocomposites absorb a tiny quantity of moisture, and the contact points between CNOs do not change significantly. However, at higher RH levels, the adsorbed water can cause a volumetric expansion of the polymer (swelling) by disrupting many of the hydrogen bonds established between its alcoholic groups (Figure 13). Consequently, the contact points between CNOs decrease rapidly, percolating pathways are strongly diminished, and CNOs–PVA-based RH sensitive layers become more resistive. Thus, for RH > 82% (for CNOs–PVA at 1:1 *w*/*w*-based sensing layer) and RH > 50% (for CNOs–PVA at 2:1 *w*/*w*-based sensing layer), one can assume that the swelling of the hydrophilic polymer is the most significant RH sensing mechanism.

The RH sensing mechanism in the case of CNO/PVA mixture sensing is complex, as supported by our experimental data. The primary RH sensing mechanism for RH < 82% (for CNOs–PVA at 1:1 *w*/*w*) and RH < 50% (for CNOs–PVA at 2:1 *w*/*w*) appears to be the interaction between water molecules and CNOs, which act as p-type semiconductors. Water molecules donate electron pairs upon contact, reducing hole concentration in the CNOs and thus decreasing conductivity. This mechanism aligns with the HSAB principle, where water molecules (hard bases) interact with CNOs (hard acids), further disrupting conductive pathways. Additionally, the swelling of the hydrophilic PVA matrix at higher RH levels plays a crucial role, increasing the distance between CNOs and diminishing percolation pathways, which significantly increases resistance. While we acknowledge the mechanism proposed by Dhonge et al. [69] involving a p-type to n-type transition in CNOs, our experimental results do not support this behavior, as no decrease in resistance was observed at high RH. Similarly, the contribution of proton conduction from water self-ionization, while plausible, appears to play a minor role in resistive RH monitoring in our study. Based on our findings, we conclude that the dominant RH sensing mechanism is the reduction in hole concentration in CNOs for lower RH values. At the same time, at higher RH levels, the swelling of the PVA matrix becomes the most significant factor.

The swelling of the PVA molecular backbones via moisture adsorption was broadly exploited in manufacturing RH sensors. Holey CNHs–PVA [59], SWCNTs–PVA [74], MWCNTs–PVA composite yarn [75], and GO–PVA are just a few examples [76]. Each of the listed nanocomposites exhibits an RH threshold (value of RH corresponding to a dramatic increase in the resistance vs. RH dependence), which can be modulated through several parameters, such as PVA polymerization degree, cross-linking degree, mutual interaction with nanocarbon material, PVA–nanocarbon material *w*/*w* ratio, etc. [75].

It is interesting to point out the significant difference between these nanocomposites and PVP/nanocarbon materials-based nanocomposites used as RH sensing layers within the architecture of resistive sensors. While all the discussed PVA-based nanocomposites exhibit the RH threshold value at high RH, CNOs-PVP nanocomposite has a RH threshold value occurring at lower RH, around 50% [77], while the CNHox-PVP nanocomposite does not show any threshold value at all [59].

The first possible explanation for this phenomenon is related to the uncommonly low value of the water vapor permeability of PVA compared with that of PVP and other hydrophilic polymers [78]. Water penetrates PVA only at high RH, and only at this point does enough water penetrate the polymers and expand their volume (swelling). In the case of CNHox-PVP, PVP’s higher water vapor permeability than PVA goes together with the mutual interaction, through hydrogen bonds between PVP and nanocarbon materials. Thus, water penetrates continuously and linearly into the matrix nanocomposite.

Extending the range of CNOs/PVA formulations could offer valuable insights into optimizing sensor performance. However, this study focused on two specific ratios, CNO:PVA 1:1 and 2:1, as an initial step to assess the feasibility of using this composite for relative humidity (RH) sensing. These ratios (50% CNO and 66.66% CNO) were selected to ensure conductivity above the percolation threshold, enabling effective sensor functionality. The results demonstrate that these formulations exhibit promising performance. While further testing is needed to optimize the sensor and explore additional formulations, the findings provide a solid foundation for future research and confirm the effectiveness of the CNO-PVA composite for RH sensing.

## 5. Conclusions

This paper presents several preliminary investigations regarding the RH detection response of a chemiresistive sensor that uses a novel sensing layer based on CNOs and PVA at 1/1 and 2:1 *w*/*w* ratios. The sensing device, including a Si/SiO_2_ substrate and gold electrodes, is obtained by drop casting the CNOs–PVA aqueous suspension on the Si/SiO_2_ substrate. The morphology and composition of the synthesized sensing layers are investigated by SEM, Raman spectroscopy, AFM, and XRD. The RH performance of the designed sensors at RT was explored by applying a continuous flow of the electric current between the interdigitated electrodes and measuring the voltage as the RH was varied from 5% to 95%. For an RH threshold below 82% (in the case of CNOs–PVA at 1/1 *w*/*w* ratio) or below 50.5% (in the case of CNOs–PVA at 2/1 *w*/*w* ratio), the resistance showed a linear variation with RH, with a moderate slope, while for RH values above these thresholds, significantly steeper slopes were measured. The change in relative resistance (ΔR/ΔRH) across different humidity steps showed that the CNOs–PVA sensor was more sensitive at humidity levels above 80%. The newly developed sensor using CNOs–PVA in a 1:1 (*w*/*w*) ratio showed response times similar to or better than those of the reference sensor. Its recovery time was about 30 s, roughly half that of the reference sensor. The decrease in the hole concentration in the CNOs in interaction with an electron donor molecule, such as water, and the swelling of the hydrophilic PVA polymer were demonstrated to be the main RH sensing mechanisms explaining the experimental results. The experimental data were analyzed and compared with the previously published RH sensing behavior of sensing layers based on CNHox–PVP, CNHox–PVA, and CNOs–PVP. The developed sensor demonstrates several competitive advantages compared to existing analogs. Unlike many conventional sensors that rely on metals or metal oxides, the proposed sensing device utilizes a carbon nano-onions (CNO) and polyvinyl alcohol (PVA) mixture, making it an environmentally friendly alternative. This metal-free composition reduces environmental impact while maintaining effective sensing capabilities. Additionally, the sensor operates on a resistive mechanism, ensuring low energy consumption, which is a critical advantage for sustainable and energy-efficient applications. The triplicate data presented for the CNO–PVA 1:1 formulation confirms the reliability of the preparation method, yielding sensors with excellent reproducibility and stability over multiple operating cycles. These results highlight the robustness and practicality of the proposed sensor, making it a promising candidate for relative humidity (RH) sensing applications.

## Figures and Tables

**Figure 1 sensors-25-03047-f001:**
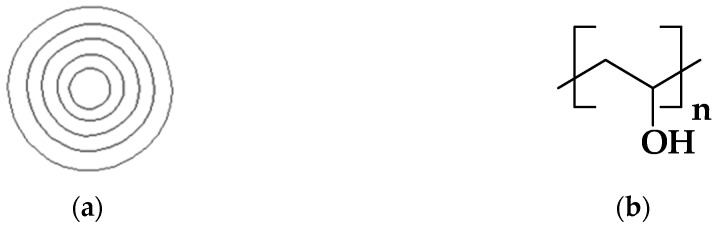
The structure of (**a**) CNOs and of (**b**) PVA.

**Figure 2 sensors-25-03047-f002:**
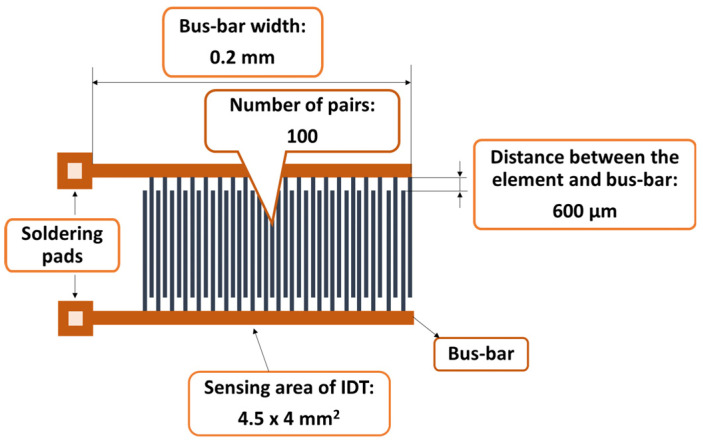
The metal stripes of the IDT.

**Figure 3 sensors-25-03047-f003:**
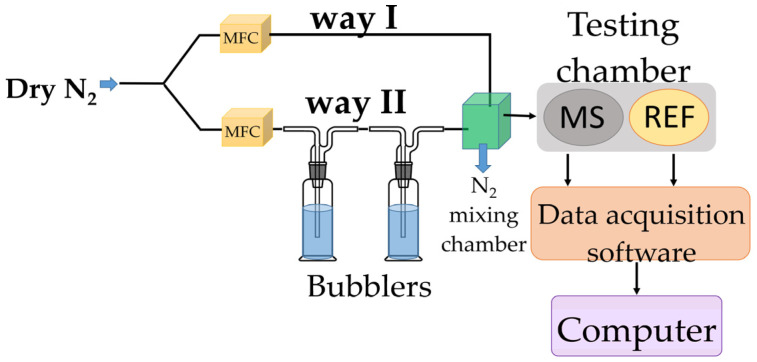
RH measurements experimental setup.

**Figure 4 sensors-25-03047-f004:**
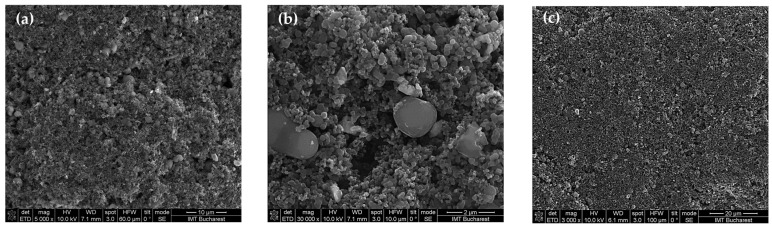
SEM of CNOs–PVA, 1:1 (*w*/*w*) layer at: (**a**) ×5000, (**b**) ×30,000, and (**c**) ×3000 magnification.

**Figure 5 sensors-25-03047-f005:**
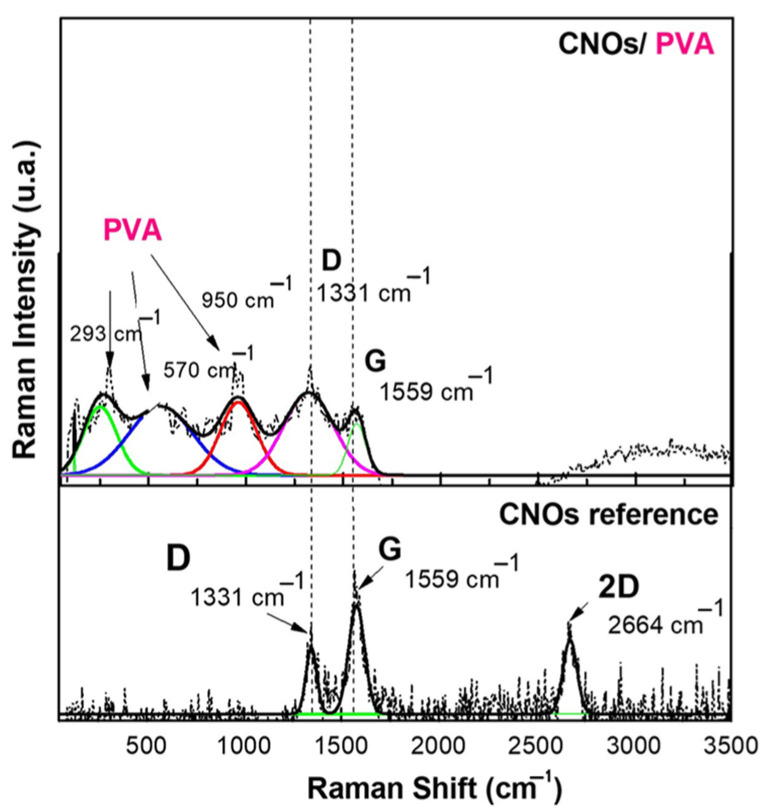
Raman spectra of a solid-state film of CNOs–PVA (1/1 *w*/*w* mass ratio) deposited on the Si substrate.

**Figure 6 sensors-25-03047-f006:**
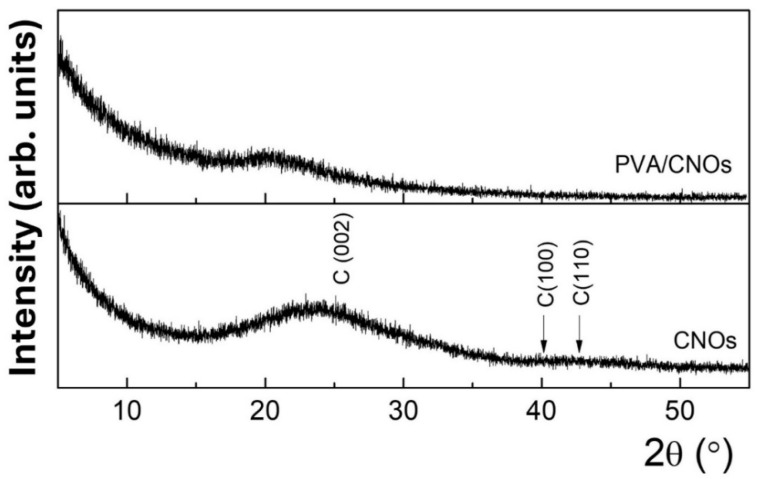
XRD pattern for CNOs–PVA 1/1 *w*/*w* ratio (above) and CNOs (below).

**Figure 7 sensors-25-03047-f007:**
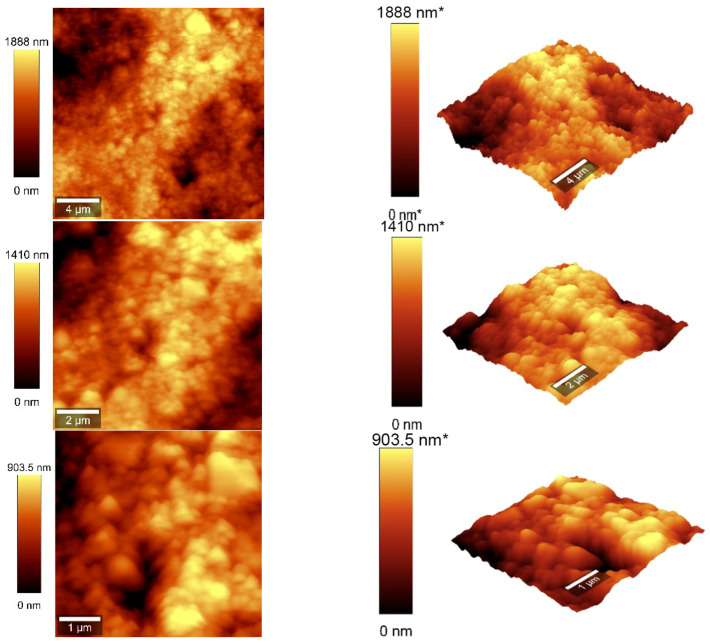
AFM images of the CNOs–PVA RH sensing layer (1/1 *w*/*w* ratio).

**Figure 8 sensors-25-03047-f008:**
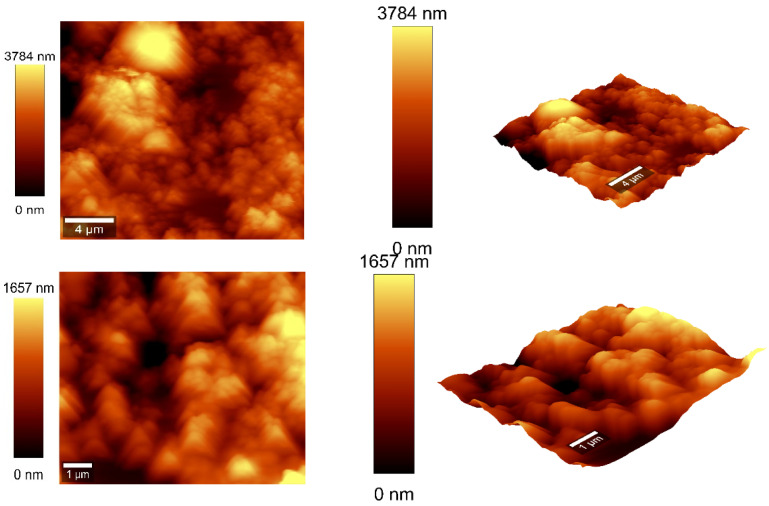
AFM images of the CNOs–PVA RH sensing layer 2:1 *w*/*w* mass ratio.

**Figure 9 sensors-25-03047-f009:**
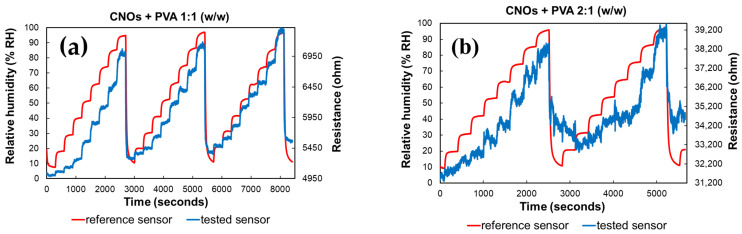
RH and resistance variation vs. time for two types of CNOs–PVA sensing layers: (**a**) 1:1 (*w*/*w*), and (**b**) 2:1 (*w*/*w*).

**Figure 10 sensors-25-03047-f010:**
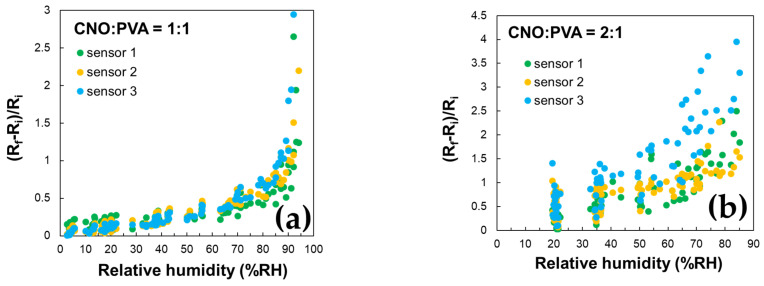
RH variation in repeated sorption–desorption cycles of three identically manufactured sensors employing the following as sensing layer: (**a**) CNOs–PVA 1:1 (*w*/*w*), and (**b**) CNOs–PVA 2:1 (*w*/*w*).

**Figure 11 sensors-25-03047-f011:**
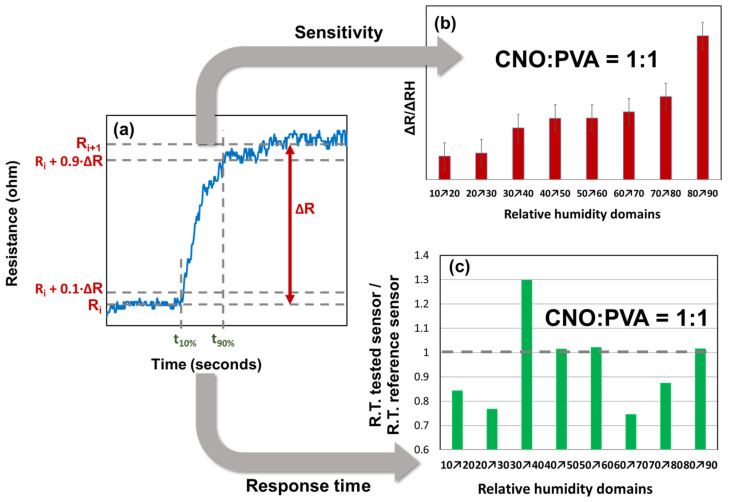
Response time and relative resistance for each RH variation step: (**a**) calculated parameters for a single RH variation step, (**b**) ΔR/ΔRH values (red bars), and (**c**) response time (green bars).

**Figure 12 sensors-25-03047-f012:**
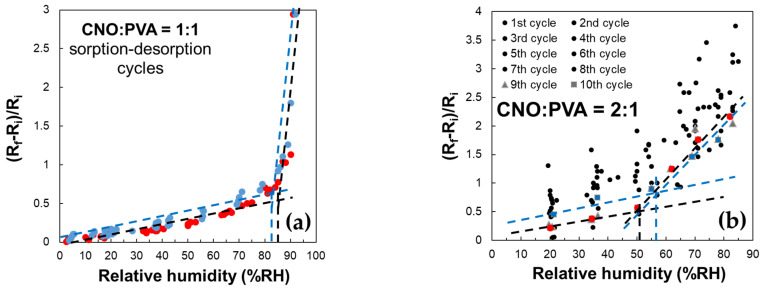
Relative variation in the resistance (Rf − Ri)/Ri as a function of RH for MS employing as sensing layer: (**a**) CNOs–PVA at 1:1 (*w*/*w*), and (**b**) CNOs–PVA at 1:1 (*w*/*w*) (sorption—red dots, desorption—blue dots).

**Figure 13 sensors-25-03047-f013:**
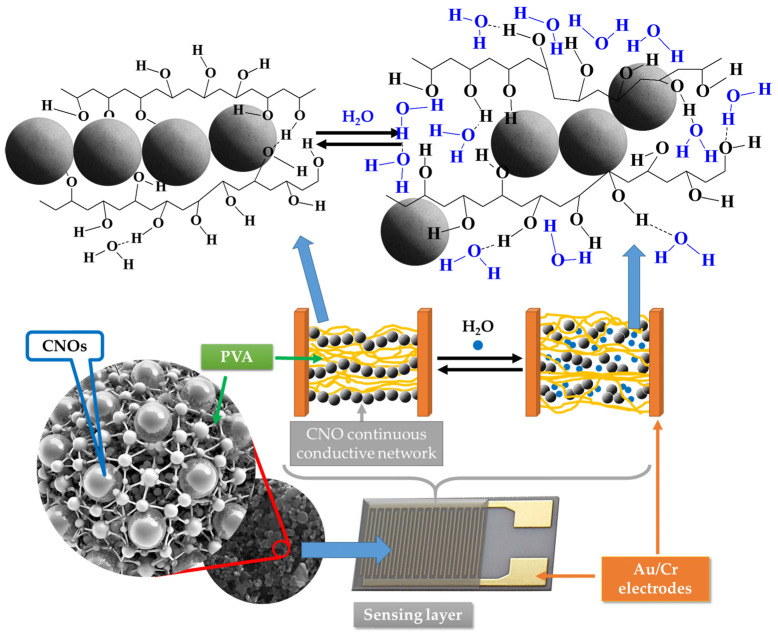
The swelling effect occurring in the PVA in contact with water molecules leads to the disruption of hydrogen bonds between hydroxyl groups of PVA and the percolating pathways of the CNOs.

**Table 1 sensors-25-03047-t001:** Nanocarbon materials-based nanocomposites and nanohybrids used as RH sensing films.

Type of Nanocomposite/Nanohybrid	Type of Sensor	Reference
Sodium hyaluronate (SH)–multi-walled carbon nanotubes (MWCNTs)	Resistive	[44]
Nitrocellulose–MWCNTs	Resistive	[45]
NaCl/OH–MWCNTs	Electrochemical	[46]
PVA–graphene flower	Capacitive	[47]
PVP-reduced graphene oxide (RGO)	Resistive	[48]
Graphene oxide (GO)–PVA	Optical	[49]
Cellulose nanocrystals–GO	Capacitive	[50]
MWCNTs–polyimide	Resistive	[51]
Single-walled carbon nanotubes (SWCNTs)–PVA filaments	Resistive	[52]
Flexible cellulose–carbon nanotubes	Strain / Stress	[53]
PVP–carbon dots	Resistive	[54]
KCl/carbon black/halloysite nanotubes	Electrochemical	[55]
Nanodiamond–cellulose nanocrystals	Surface acoustic wave (SAW)	[56]
Flake-like nanodiamond–chitosan composite	SAW	[57]
GO–oxidized carbon nanohorns (CNHox)–PVP	Resistive	[58]
CNHox–PVP	Resistive	[59]
CNHox–PVA	Resistive	[59]

## Data Availability

The original contributions presented in this study are included in the article. Further inquiries can be directed to the corresponding author(s).

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
