# Peer review of "Carbon Nano-Onions–Polyvinyl Alcohol Nanocomposite for Resistive Monitoring of Relative Humidity"

_sensors, 2025, doi:10.3390/s25103047_

Round 1
Reviewer 1 Report
Comments and Suggestions for Authors
In this manuscript, the authors investigated Carbon nano-onions-polyvinyl alcohol nanocomposite as sensing film for resistive monitoring of relative humidity. The results are valuable to the readership in this area. The novelty and significance of this work is qualified to be published in this journal. I recommend this manuscript can be accepted for publication after minor revisions.
- “The doubling of this ratio indicates a notable increase in structural defects and disorder, which is attributed to the incorporation of PVA within the composite matrix.” in Page 6, please explain the mechanism and support it with relevant articles.
- For better understanding, please adjust the scale of Figure 4 or provide a clearer picture.
- “The CNOs concentration is above the percolation threshold of the nanocarbon material in the hydrophilic matrix of PVA. Therefore, electrically percolating paths between the two metal electrodes of the proposed RH sensor make it easy to measure electrical resistance.” in Page 11, please explain the mechanism.
- There should be Spaces between symbol and numbers, such as 50‐60℃, please check the full text.
- “The HSAB theory principle states that between hard acids and hard bases chemical interaction is possible. Therefore, the chemical interaction between CNOs and water seems to be feasible.” in Page 12, please explain the HSAB principle in detail.
- The number of references in recent years is too small, it is recommended to add the latest research on advanced materials and high-performance sensors in the introduction, such as Journal of Colloid and Interface Science, v 606, p 261-271, 2022, Talanta, v 182, p 324-332, 2018, Rare Metals, v 41, n 9, p 3117-3128, 2022, Chemical Engineering Journal, v 495, p 153676, 2024, Nano Energy, v 127, p 109753, 2024, ACS Applied Materials & Interfaces, v 15, n 4, p 5811-5821, 2023.
Author Response
Dear Reviewer,
Thank you for your valuable observations, which are helping us improve the manuscript. Below, you will find a point-by-point response to the highlighted issues.
- “The doubling of this ratio indicates a notable increase in structural defects and disorder, which is attributed to the incorporation of PVA within the composite matrix.” in Page 6, please explain the mechanism and support it with relevant articles.
Thank you for the observation. A clearer paragraph was added in the manuscript.
“The D and G bands (ID/IG) intensity ratio in Raman spectra provides a quantitative measure of defect density and structural disorder in carbon-based materials. For pristine carbon nano onions (CNOs), this ratio is ID/IG = 0.558, while for the CNO/PVA composite, it increases significantly to ID/IG = 1.126. This doubling of the ratio indicates a substantial rise in structural defects and disorder, which can be attributed to the polyvinyl alcohol (PVA) incorporation into the composite matrix. The interaction between PVA and CNOs, involving the formation of covalent and non-covalent bonds, alters the material's structural arrangement and Raman scattering intensity. These interactions are critical for the sensing behavior of the composite in resistive sensors”
References “Rosenkranz, A., Freeman, L., Fleischmann, S., Lasserre, F., Fainman, Y. and Talke, F.E., 2018. Tip-enhanced Raman spectroscopy studies of nanodiamonds and carbon onions. Carbon, 132, pp.495-502.” and “Bokova-Sirosh, S.N., Pershina, A.V., Kuznetsov, V.L., Ishchenko, A.V., Moseenkov, S.I., Orekhov, A.S. and Obraztsova, E.D., 2013. Raman spectra for characterization of onion-like carbon. Journal of nanoelectronics and optoelectronics, 8(1), pp.106-109.” were cited in the manuscript.
- For better understanding, please adjust the scale of Figure 4 or provide a clearer picture.
Thank you. The images from figure 4 were replaced with images with better contrast and better resolution.
- “The CNOs concentration is above the percolation threshold of the nanocarbon material in the hydrophilic matrix of PVA. Therefore, electrically percolating paths between the two metal electrodes of the proposed RH sensor make it easy to measure electrical resistance.” in Page 11, please explain the mechanism.
The highlighted text was replaced in the manuscript with the following paragraph to show a clearer image of the phenomenon: “For both sensing films (CNOs-PVA 2/1 and 1/1 (w/w) ratio, respectively), the concentration of CNOs is above the percolation threshold of the nanocarbon material within the hydrophilic PVA matrix. The percolation threshold is the critical concentration of conductive nanomaterials needed to form continuous, electrically conductive pathways throughout the matrix. When the CNOs concentration is above this threshold, electrically percolating paths are established between the two metal electrodes of the proposed RH sensor, allowing the device to measure changes in electrical resistance effectively. Below the percolation threshold, these conductive pathways are not formed, and the sensing device is unable to detect any changes in resistance, regardless of the humidity level in the testing environment. This is because the lack of interconnected conductive paths prevents the flow of electrical current, rendering the device insensitive to variations in relative humidity.”
- There should be Spaces between symbol and numbers, such as 50‐60℃, please check the full text.
The modification was made as requested. Thank you for the observation.
- “The HSAB theory principle states that between hard acids and hard bases chemical interaction is possible. Therefore, the chemical interaction between CNOs and water seems to be feasible.” in Page 12, please explain the HSAB principle in detail.
The following paragraph was added in the manuscript to explain the HSAB principle in detail for the proposed sensing device: “Another RH sensing mechanism is related to the interaction between CNOs and water molecules, which can be explained using the hard-soft acid-base (HSAB) princi-ple [63]. This concept is widely applied in nanotechnology to explain the stability of compounds, reaction mechanisms, and potential interactions [64]. Pristine CNOs pos-sess positively charged carriers, which can be compared to hard acids, while water molecules, due to their electron pairs, behave as hard bases. According to the HSAB principle, chemical interactions are likely to occur between hard acids and hard bases. In this context, water molecules tend to gather near the nanocarbon material (CNOs), leading to a chemical interaction that influences the sensing mechanism.
As the relative humidity increases, water molecules accumulate in the vicinity of the CNOs and interact with them. This interaction, combined with the swelling of the hydrophilic PVA matrix, increases electrical resistance. The swelling of the PVA matrix increases the distance between conductive CNOs, while the interaction with water molecules further disrupts the conductive pathways within the material. Together, these effects reduce the overall conductivity of the sensing film, translating the in-crease in humidity into a measurable increase in resistance.”
- The number of references in recent years is too small, it is recommended to add the latest research on advanced materials and high-performance sensors in the introduction, such as Journal of Colloid and Interface Science, v 606, p 261-271, 2022, Talanta, v 182, p 324-332, 2018, Rare Metals, v 41, n 9, p 3117-3128, 2022, Chemical Engineering Journal, v 495, p 153676, 2024, Nano Energy, v 127, p 109753, 2024, ACS Applied Materials & Interfaces, v 15, n 4, p 5811-5821, 2023.
The results presented in the references “Wang, D., Zhang, D., Yang, Y., Mi, Q., Zhang, J. and Yu, L., 2021. Multifunctional latex/polytetrafluoroethylene-based triboelectric nanogenerator for self-powered organ-like MXene/metal–organic framework-derived CuO nanohybrid ammonia sensor. ACS nano, 15(2), pp.2911-2919.”, and “Shao, X., Zhang, D., Tang, M., Zhang, H., Wang, Z., Jia, P. and Zhai, J., 2024. Amorphous Ag catalytic layer-SnO2 sensitive layer-graphite carbon nitride electron supply layer synergy-enhanced hydrogen gas sensor. Chemical Engineering Journal, 495, p.153676.” supports very well the HSAB theory application in the sensing mechanism.
Thank you!
Reviewer 2 Report
Comments and Suggestions for Authors
The article "Carbon nano-onions-polyvinyl alcohol nanocomposite as sensing film for resistive monitoring of relative humidity" discusses the development of a chemoresistive humidity sensor, which uses a nanocomposite based on carbon nanolubes (CNOs) and polyvinyl alcohol (PVA) as a sensitive layer. The authors investigated two ratios of CNOs and PVA (1:1 and 2:1) and studied their effect on the characteristics of the humidity sensor. The morphology and composition of the synthesizer sensing layers are investigated by SEM, Raman spectroscopy, ARM and XRD. The work is of interest to the scientific community involved in the development of humidity sensors., Below are the comments that will help the authors to finalize the manuscript as follows:
- Expand paragraph 3.1 to include more details about the observed surface characteristics, an explanation of the heterogeneity, and a discussion of how this surface topography may affect the characteristics of the humidity sensor;
- It is recommended to move Figures 5,6,7,8 and 12 after the paragraphs in which they are first mentioned;
- Improve the quality of drawings;
- Provide data on the hysteresis and stability of the sensor in order to assess its suitability for practical use;
- Only two CNOs/PVA ratios (1:1 and 2:1) have been studied. Extending the range of formulations could help optimize sensor performance;
- It is necessary to conduct research on the long-term stability of the sensor and suggest ways to improve it;
- In the conclusion of the article, the competitive advantages of the developed sensor should be more clearly formulated in comparison with existing analogues, based on the results obtained.
Thus, the article is recommended for publication, but after completion.
Author Response
Dear Reviewer,
Thank you for your valuable observations, which have greatly contributed to improving our manuscript. Below, we provide a point-by-point response to the highlighted issues to enhance the presentation of our work.:
- Expand paragraph 3.1 to include more details about the observed surface characteristics, an explanation of the heterogeneity, and a discussion of how this surface topography may affect the characteristics of the humidity sensor;
Thank you for the observation. The paragraph at section 3.1 was replaced with the following paragraph:”The surface topography of the sensing films based on the CNOs–PVA (carbon nano-onions–polyvinyl alcohol) nanocomposite was characterized using scanning electron microscopy (SEM). Figure 4 presents the morphology of the sensing layer prepared with a CNO/PVA ratio of 1:1 (w/w). As shown in Figure 4a, the surface of the film is generally homogeneous, with a good distribution of nanocarbon materials throughout the polymer matrix. However, closer inspection (Figure 4b) reveals the presence of localized agglomerations, forming isolated islands of CNO nanoparticles as well as distinct PVA polymer clusters.
The CNOs were initially dispersed into a transparent water-based PVA solution and subjected to intensive ultrasonic treatment for 6 hours at a controlled temperature of 50–60 °C. These preparation conditions were sufficient to promote a relatively good dispersion of the CNOs within the PVA matrix. After deposition, the sensing films were annealed under moderate vacuum conditions at a moderate temperature for 4 hours to minimize nanoparticle agglomeration. Despite these careful processing steps, some minor aggregation of nanoparticles was still observed, as expected in such nanocomposites.
Overall, the surface morphology indicates an acceptable level of homogeneity, which is critical for ensuring reliable sensor performance. The good film continuity and dispersion achieved, as shown in Figure 4a, confirm that the prepared CNOs–PVA sensing layer is suitable for use in the intended resistive humidity sensor applications.”
- It is recommended to move Figures 5,6,7,8 and 12 after the paragraphs in which they are first mentioned;
The modifications were made as recommended. Thank you.
- Improve the quality of drawings;
Figure 2, 3, 14 were replaced with iages wuth better resolution as indicated.
- Provide data on the hysteresis and stability of the sensor in order to assess its suitability for practical use;
The paragraph:
“Significant differences in resistance variation were observed when comparing the two RH sensing layers. The sensor employing CNOs-PVA at 1:1 (w/w) exhibited resistance values between 5 and 7.7 kΩ when varying RH in the 5%–95% range, whereas the structure using CNOs-PVA at 2:1 (w/w) showed a much higher resistance variation, ranging from 31 to 40 kΩ, under the same RH variation conditions (Figure 9).”
was replaced with the following paragraph to emphasize sensor CNO:PVA 1:1 suitability for practical use:
“Significant differences in resistance variation were observed between the two RH sensing layers. The sensor utilizing a CNOs-PVA ratio of 1:1 (w/w) demonstrated re-sistance values ranging from 5 to 7.7 kΩ when exposed to relative humidity (RH) levels between 5% and 95%. In contrast, the sensor with a CNOs-PVA ratio of 2:1 (w/w) ex-hibited much higher resistance values, varying between 31 and 40 kΩ under the same RH conditions (Figure 9). Although a higher concentration of conductive CNOs is ex-pected to enhance electrical conductivity, the experimental results indicate a resistance increase for the CNO:PVA (2:1 w/w) sensing layer. This unexpected behavior can be attributed to the agglomeration of nanoparticles at higher CNO concentrations, which disrupts the formation of a continuous conductive network. Instead, isolated clusters with poor electrical connectivity are formed. Furthermore, an excessive CNO content compromises the continuity of the hydrophilic PVA matrix, impairing charge transport and reducing humidity sensing performance. Contact resistance between inadequately connected CNO clusters may also contribute to the observed increase in resistance.
The hysteresis curves presented in Figure 9 further highlight the differences be-tween the two sensors. The sensor with a CNO:PVA ratio of 1:1 stabilizes after the first operating cycle and, by the third cycle, exhibits a response comparable to that of a commercial sensor. In contrast, the sensor with a ratio of 2:1 fails to stabilize even after the first two cycles, indicating that higher CNO concentrations are unsuitable for use in sensing devices.
These findings are supported by the data in Figure 10, which demonstrate the ex-cellent stability of the CNO:PVA 1:1 sensor over multiple operating cycles.”
- Only two CNOs/PVA ratios (1:1 and 2:1) have been studied. Extending the range of formulations could help optimize sensor performance;
Thank you for your valuable comment. We acknowledge that extending the range of CNOs/PVA formulations could provide further insights into optimizing sensor performance. However, in this study, we focused on two specific ratios, CNO:PVA 1:1 and 2:1, as a starting point to evaluate the feasibility of using this composite for relative humidity (RH) sensing.
The selected ratios (50% CNO and 66.66% CNO) were intentionally chosen to be above the percolation threshold, ensuring sufficient conductivity for sensor functionality. The primary objective was to determine whether the mixture would be suitable for RH detection. We are pleased to report that the selected formulations demonstrated promising performance in this regard.
Although further tests are necessary to optimize the sensor and fully explore the range of formulations, we believe the presented results provide a strong foundation for future studies. Our findings demonstrate that the CNO:PVA composite is effective for RH sensing, and we aim to investigate additional ratios and establish the percolation threshold in subsequent phases of research.
The paragraph bellow was added at the end of Section 4:
“Extending the range of CNOs/PVA formulations could offer valuable insights into optimizing sensor performance. However, this study focused on two specific ratios, CNO:PVA 1:1 and 2:1, as an initial step to assess the feasibility of using this composite for relative humidity (RH) sensing. These ratios (50% CNO and 66.66% CNO) were selected to ensure conductivity above the percolation threshold, enabling effective sensor functionality. The results demonstrate that these formulations exhibit promising performance. While further testing is needed to optimize the sensor and explore additional formulations, the findings provide a solid foundation for future research and confirm the effectiveness of the CNO:PVA composite for RH sensing.”
- It is necessary to conduct research on the long-term stability of the sensor and suggest ways to improve it;
The paragraph bellow was added at the end of Section 4:
“Extending the range of CNOs/PVA formulations could offer valuable insights into optimizing sensor performance. However, this study focused on two specific ratios, CNO:PVA 1:1 and 2:1, as an initial step to assess the feasibility of using this composite for relative humidity (RH) sensing. These ratios (50% CNO and 66.66% CNO) were selected to ensure conductivity above the percolation threshold, enabling effective sensor functionality. The results demonstrate that these formulations exhibit promising performance. While further testing is needed to optimize the sensor and explore additional formulations, the findings provide a solid foundation for future research and confirm the effectiveness of the CNO:PVA composite for RH sensing.”
- In the conclusion of the article, the competitive advantages of the developed sensor should be more clearly formulated in comparison with existing analogues, based on the results obtained.
The paragraph was addedin the conclusion section: “The developed sensor demonstrates several competitive advantages compared to existing analogues. Unlike many conventional sensors that rely on metals or metal oxides, the proposed sensing device utilizes a carbon nano-onions (CNO) and polyvinyl alcohol (PVA) mixture, making it an environmentally friendly alternative. This metal-free composition reduces environmental impact while maintaining effective sensing capabilities. Additionally, the sensor operates on a resistive mechanism, ensuring low energy consumption, which is a critical advantage for sustainable and energy-efficient applications. The triplicate data presented for the CNO:PVA 1:1 formulation confirms the reliability of the preparation method, yielding sensors with excellent reproducibility and stability over multiple operating cycles. These results highlight the robustness and practicality of the proposed sensor, making it a promising candidate for relative humidity (RH) sensing applications.”
Thank you for your time and your observations that helped us to improve the presentation of our work.
Reviewer 3 Report
Comments and Suggestions for Authors
This manuscript (sensors-3605066) reported a humidity sensor utilizing CNOs‐PVA sensing layers. The humidity sensing results are acceptable, but there are many issues in research motivation, results and discussion, and writing. The manuscript may be accepted after correcting the following issues:
- The title is too complicated. It is suggested that the authors consider, for example, Carbon Nanoparticles-Polyvinyl Alcohol Nanocomposite for Humidity Sensor. Except for prepositions, articles, and conjunctions, the first letters of all words need to be capitalized for title and subtitles.
- Why are carbon nanoparticles referred to as Carbon nano-onions?
- Si/SiO2 substrate: The numbers in the chemical formula require subscripts, including references.
- The abbreviations that first appear need to be spelled in full. Check for similar issues. …nanohorns‐PVA and CNOs‐polyvinylpyrrolidone (PVP). The PVP here does not require complete spelling.
- Introduction: (1) It is recommended to discuss the advantages and disadvantages of different types of humidity sensors and highlight the significance of this work of resistance humidity sensor (such as resistance, capacitance, impedance, frequency, optics, self-powered (TENG, ion gradient, and electrochemistry)), may referring to ACS Nano 2024, 18, 34158-34170. (2) “…metal oxide semiconductors [7, 8],”. Reference [8] is actually a moisture resistant gas sensor, not a humidity sensor. It is recommended to cite metal oxide and ceramic humidity sensors (such as halloysite nanotubes, attapulgite, and sepiolite nanofibers). (3) Carbon material humidity sensors typically exhibit positive humidity sensitive resistance characteristics, meaning that as humidity increases, the resistance of carbon-based humidity sensors will rise. Especially for carbon/organic composite materials, due to their hygroscopic swelling blocking effect. Suggest discussing the research progress of carbon-based positive humidity sensors, such as combining reference [14].
- Table 1 can be placed in the Results and Discussion section, in conjunction with the humidity sensing performance of the humidity sensor.
- How does Figure 2 generate different RHs? The logic of the gas path inlet/outlet is incorrect. Suggest referring to relevant references.
- The scale bar in Figure 4 is unclear, it is recommended to add it again.
- The sensitivity calculation formula needs to be limited to the linear response range.
- The humidity sensing mechanism needs to be reconsidered. For carbon/organic materials, moisture absorption and expansion dominate (referring to Fig. 10 in ref 14). The proposed conductive mechanisms are inaccurate.
- Check the format of the references, such as abbreviating the journal name.
- Check and improve English writing.
Check and improve English writing.
Author Response
Dear Reviewer,
Thank you for your valuable observations, which have greatly contributed to improving our manuscript. Below, we provide a point-by-point response to the highlighted issues to enhance the presentation of our work.:
- The title is too complicated. It is suggested that the authors consider, for example, Carbon Nanoparticles-Polyvinyl Alcohol Nanocomposite for Humidity Sensor. Except for prepositions, articles, and conjunctions, the first letters of all words need to be capitalized for title and subtitles.
The title was modified. The new title is: “Carbon nano-onions-polyvinyl alcohol nanocomposite for resistive monitoring of relative humidity” to avoid redundancy. Thank you.
- Why are carbon nanoparticles referred to as Carbon nano-onions?
Thank you for the observation. Indeed, the introduction section should indicate information about the carbon nano-onions used in the study.
The paragraph was added in the introduction section:
“Carbon nano-onions are multi-layered, concentric, spherical structures composed of graphene-like shells. They resemble "onions," with each layer being a curved carbon sheet, typically arranged around a central core (often a fullerene or amorphous carbon). CNOs are typically in the nanometer range, with diameters ranging from a few nanometers to tens of nanometers, depending on the synthesis method and application. Carbon nano-onions (CNOs) are a distinct and highly specialized form of carbon nanomaterial with unique structural and functional properties.”
AS a consequence, referring to them as "carbon nanoparticles" would be inaccurate and misleading, as it fails to capture their well-defined concentric structure and exceptional performance characteristics. Using "carbon nanoparticles" to describe CNOs would overlook the structural and functional advantages that make them suitable for specific applications, such as resistive humidity sensors.
- Si/SiO2 substrate: The numbers in the chemical formula require subscripts, including references.
Thank you for the observation. The corrections were made in the manuscript.
- The abbreviations that first appear need to be spelled in full. Check for similar issues. …nanohorns‐PVA and CNOs‐polyvinylpyrrolidone (PVP). The PVP here does not require complete spelling.
The corrections were done. Thank you.
- Introduction:
(1) It is recommended to discuss the advantages and disadvantages of different types of humidity sensors and highlight the significance of this work of resistance humidity sensor (such as resistance, capacitance, impedance, frequency, optics, self-powered (TENG, ion gradient, and electrochemistry)), may referring to ACS Nano 2024, 18, 34158-34170.
(2) “…metal oxide semiconductors [7, 8],”. Reference [8] is actually a moisture resistant gas sensor, not a humidity sensor. It is recommended to cite metal oxide and ceramic humidity sensors (such as halloysite nanotubes, attapulgite, and sepiolite nanofibers).
(3) Carbon material humidity sensors typically exhibit positive humidity sensitive resistance characteristics, meaning that as humidity increases, the resistance of carbon-based humidity sensors will rise. Especially for carbon/organic composite materials, due to their hygroscopic swelling blocking effect. Suggest discussing the research progress of carbon-based positive humidity sensors, such as combining reference [14].
Thank you for the suggestions. The Introduction section was revised accordingly.
To better contextualize the significance of this work, we have included a discussion on the advantages and disadvantages of various types of humidity sensors, including resistance, capacitance, impedance, frequency, optics, and self-powered sensors.
“These sensor types exhibit unique strengths and limitations depending on their application scenarios. Among these, resistance-based humidity sensors, such as the one developed in this work, stand out due to their simplicity, cost-effectiveness, and high sensitivity, making them suitable for a wide range of applications.”
Additionally, we have corrected the citation for reference [8], as it describes a moisture-resistant gas sensor rather than a humidity sensor. We have replaced it with relevant references on metal oxide humidity sensors, which are widely recognized for their excellent stability and sensitivity in humidity sensing applications.
Furthermore, we have expanded the discussion on carbon-based humidity sensors, which typically exhibit positive resistance characteristics due to the hygroscopic swelling effect of carbon/organic composite materials. This phenomenon, which causes an increase in resistance with rising humidity, is particularly significant for carbon/organic materials, as their swelling disrupts conductive pathways. Recent progress in carbon-based positive humidity sensors, including studies such as reference [14], highlights the potential of these materials in developing highly sensitive and reliable humidity sensors.
- Table 1 can be placed in the Results and Discussion section, in conjunction with the humidity sensing performance of the humidity sensor.
In our opinion, Table 1 effectively supports the discussion, highlighting the sensing capabilities of the nanocomposites and nanohybrids composed of polymers and nanocarbon materials used as RH sensing films. The discussion section focuses on the relationship between the proposed solution and the sensing mechanism.
- How does Figure 2 generate different RHs? The logic of the gas path inlet/outlet is incorrect. Suggest referring to relevant references.
You are probably referring to Figure 3. Indeed, it was an error in the image that was corrected. The mixing chamber collects the wet and dry nitrogen from the two ways controlled by the mass flow controllers.
A paragraph that describes the experimental setup was introduced in the Section 2.2:
“Relative humidity sensing measurements were performed using a specialized testing bench (Figure 3). To vary the relative humidity (RH) in the testing chamber, dry nitrogen was passed through two containers in series containing deionized water. The humidity level was adjusted by mixing dry nitrogen with the humidified nitrogen in different ratios. In the mixing chamber (indicated by the green square in Figure 3), the gases from both paths formed a homogeneous mixture, which was then directed into the testing chamber. The chamber housed two sensors: our resistive sensing structure (referred to as MS – measured sensor), which uses GO/CNHox/PVP as the sensing layer, and a commercially available capacitive relative humidity sensor (referred to as REF – reference sensor). The commercial sensor was used to validate the relative humidity levels indicated by the mass flow controller (MFC) system. Both sensors were positioned close to each other and near the gas inlet, ensuring identical exposure to the gas flow (dry nitrogen) and providing consistent experimental conditions for reliable conclusions [54,55].”
Thank you.
- The scale bar in Figure 4 is unclear; it is recommended to add it again.
Images from figure 4 were replaced with images of better resolution. Thank you.
- The sensitivity calculation formula needs to be limited to the linear response range.
The tested sensors exhibited an inflection point in the humidity measurements, indicating the combined effects of various interactions between the sensing CNO/PVA layer and water molecules. The sensitivity calculation (Figure 11b) demonstrates that at higher RH values, the resistance increases rapidly. Additionally, calculating sensitivity across the entire %RH range provides valuable information to support the mechanistic model proposed in the discussion section.
- The humidity sensing mechanism needs to be reconsidered. For carbon/organic materials, moisture absorption and expansion dominate (referring to Fig. 10 in ref 14). The proposed conductive mechanisms are inaccurate.
We appreciate the reviewer’s insightful feedback regarding the proposed humidity sensing mechanisms. We agree that moisture absorption and expansion are dominant factors for carbon/organic materials, as highlighted in the manuscript.
“…. .one can assume that the swelling of the hydrophilic polymer is the most significant RH sensing mechanism.”
The paragraph bellow was added in the discussion section to clarify and expand on these mechanisms in light of the reviewer’s comments.
However, we believe that the RH sensing mechanism in our case is more complex, as supported by our experimental data showing an inflection point at specific RH values. The primary RH sensing mechanism for RH < 82% (for CNOs-PVA at 1:1 w/w) and RH < 50% (for CNOs-PVA at 2:1 w/w) appears to be the interaction between water molecules and CNOs, which act as p-type semiconductors. Water molecules donate electron pairs upon contact, reducing hole concentration in the CNOs and thus decreasing conductivity. This mechanism aligns with the HSAB principle, where water molecules (hard bases) interact with CNOs (hard acids), further disrupting conductive pathways. Additionally, the swelling of the hydrophilic PVA matrix at higher RH levels plays a crucial role, increasing the distance between CNOs and diminishing percolation pathways, which significantly increases resistance. While we acknowledge the mechanism proposed by Dhonge et al. [65], involving a p-type to n-type transition in CNOs, our experimental results do not support this behavior, as no decrease in resistance was observed at high RH. Similarly, the contribution of proton conduction from water self-ionization, while plausible, appears to play a minor role in resistive RH monitoring in our study. Based on our findings, we conclude that the dominant RH sensing mechanism is the reduction of hole concentration in CNOs for lower RH values, while at higher RH levels, the swelling of the PVA matrix becomes the most significant factor.
- Check the format of the references, such as abbreviating the journal name.
Corrections were made as indicated.
- Check and improve English writing.
An extensive revision of English writing was made.
Thank you.
Round 2
Reviewer 3 Report
Comments and Suggestions for Authors
The response and revised draft are unsatisfactory.
- Please provide a separate response PDF file instead of just text.
- Please pay attention to responding to each question, especially if some questions cover multiple sub questions.
- Please explain any unresolved issues in the revised manuscript, rather than simply responding to the reviewers, as these issues may also be of concern to future readers.
- Title: Except for prepositions, articles, and conjunctions, the first letters of all words need to be capitalized for title and subtitles.
- “Carbon nano-onions are multi-layered, concentric, spherical structures composed of graphene-like shells”. carbon nano-onions? The onion structure cannot be seen from the SEM morphology image in Figure 4. How to prove the morphology structure? Either delete the statement or provide more reliable evidence, such as TEM morphology.
- The types of humidity sensors in Table 1 are not complete. For example, electrochemical humidity sensors, ion gradients, frictional humidity sensors, QCM humidity sensors, AC impedance humidity sensors, etc. In addition, Table 1 also does not cover widely used oxide and ceramic humidity sensitive materials.
- “Among these, resistance-based humidity sensors, such as the one developed in this work, stand out due to their simplicity, cost-effectiveness, and high sensitivity, making them suitable for a wide range of applications.”. Compared to other sensors, capacitive and impedance humidity sensors also have simple structures and are commercially available
- 3: Why does the author need two water bottles? How does the water vapor from the second gas cylinder come out?
- 11a: In the field of humidity sensors, the response time ranges from 0% to 90% of the response, rather than 10% to 90% (J. Mater. Chem. A, 2024, 12, 27157–27179).
- For combined images, each small image number needs to be embedded into the image.
- The numbers in the chemical formula require subscripts, including references, like Cs2SnCl6 perovskites.
- 4. AFM measurements …Atomic force microscopy (AFM)…The abbreviations that first appear need to be spelled in full. Check for similar issues.
- References list: Most of references are out of date. It is suggested to cite recent 2-3 years.
- Check and improve English writing.
Check and improve English writing.
Author Response
Dear Reviewer,
Thank you for your valuable feedback. We sincerely appreciate your effort in identifying potential flaws in the study's description, which has greatly helped us improve the manuscript. In the attached file, you will find our point-by-point response to the issues raised.
Thank you!

Round 3
Reviewer 3 Report
Comments and Suggestions for Authors
The reviewer's concerns have been properly addressed and it is acceptable.